# Determining Association between Lung Cancer Mortality Worldwide and Risk Factors Using Fuzzy Inference Modeling and Random Forest Modeling

**DOI:** 10.3390/ijerph192114161

**Published:** 2022-10-29

**Authors:** Xiu Wu, Blanchard-Boehm Denise, F.Benjamin Zhan, Jinting Zhang

**Affiliations:** 1Department of Geography and Environmental Studies, Texas State University, San Marcos, TX 78666, USA; 2School of Resource and Environmental Science, Wuhan University, Wuhan 430070, China

**Keywords:** lung cancer mortality, lung cancer mortality-non-smoking factors, health, fuzzy inference modeling, random forest modelling

## Abstract

Lung cancer remains the leading cause for cancer mortality worldwide. While it is well-known that smoking is an avoidable high-risk factor for lung cancer, it is necessary to identify the extent to which other modified risk factors might further affect the cell’s genetic predisposition for lung cancer susceptibility, and the spreading of carcinogens in various geographical zones. This study aims to examine the association between lung cancer mortality (LCM) and major risk factors. We used Fuzzy Inference Modeling (FIM) and Random Forest Modeling (RFM) approaches to analyze LCM and its possible links to 30 risk factors in 100 countries over the period from 2006 to 2016. Analysis results suggest that in addition to smoking, low physical activity, child wasting, low birth weight due to short gestation, iron deficiency, diet low in nuts and seeds, vitamin A deficiency, low bone mineral density, air pollution, and a diet high in sodium are potential risk factors associated with LCM. This study demonstrates the usefulness of two approaches for multi-factor analysis of determining risk factors associated with cancer mortality.

## 1. Introduction

Lung cancer is defined by the World Health Organization’s (WHO), “International Classification of Diseases”, as a malignant neoplasm (tumor) of the trachea, bronchus, and/or lungs. About 98 to 99 percent of lung cancers are carcinomas, thus, this disease is often referred to as lung carcinoma [1]. Based on information from WHO’s GLOBOCAN 2020 estimates of cancer incidence and mortality, lung cancer remains the leading cause of cancer-related deaths in the world, with an estimated 1.8 million deaths (18 percent) worldwide. Lung cancer has an incidence ratio of 1 out of 10 of newly diagnosed cancer cases, and a mortality rate of 1 out of 5 deaths worldwide [2,3].

More recently, the medical community continues to report that, worldwide, approximately 85 percent of lung cancer deaths are due to long-term smoking [4]. The etiology in this regard has been well-established over decades of research—it is the remaining 15 percent of deaths of non-smokers that has left the research community puzzled for the past six decades. Potential risk factors for lung cancer in this regard include passive smoking, radon gas, asbestos, aerosols from mining and metal processing, combustion (indoor emissions, exhaust, and petroleum processing), ionizing radiation, toxic gasses, rubber production, and silica processing. Further, Turner and colleagues also observe that, in more recent times, emissions from industry, power generation, transportation, and domestic burning, exceed considerably the WHO’s health-based air-quality guidelines and subject the world’s population to unsafe levels of air pollution. They suggest that outdoor air pollution is an urgent worldwide public health challenge, particularly in relation to cancers [5].

The question of how to compare the impacts of human diet, habits, and environmental risk factors on lung cancer is still a challenging task. To fill these knowledge gaps, we used fuzzy inference of weight and a Random Forest Tree (RFM) model to assess the weights of the aforementioned variables in association with lung cancer mortality (LCM). The common characteristics or determinants are revealed in the comparison of the two methods in the analysis. For decades, previous studies have mainly focused on the genetic and biological aspects of lung cancer, as well as the efficacy of medical surgeries and treatment protocols for lung carcinomas [6,7,8]. Moreover, the relationship between demographic and socioeconomic variables—gender, age structure, race, income, diet and food access, level of a country’s development—have been extensively researched [9,10,11,12,13,14]. However, in more recent times, researchers such as Turner and colleagues have been focusing on the association between environmental risk factors and lung cancer. Yet, their focus is limited to one aspect of lung cancer’s incidence and mortality (i.e., indoor and outdoor pollution). These diverse perspectives, no matter the focus, tend to be localized in one specific geographical region and, findings are hence limited to specific geographic regions [15,16,17,18]. Therefore, there is a need to investigate if other factors such as human diet and habits, and physical and mental health variables are associated with LCM. In addition, this study utilizes machine learning analysis and fuzzy inference rather than multivariate statistics to explore the association between these factors and LCM.

## 2. Materials and Methods

### 2.1. Data

We collected lung cancer mortality data and mortality data associated with 30 risk factors in 100 countries during the period from 2006 to 2016, from open datasets published by the Global Burden of Disease (https://vizhub.healthdata.org/gbd-results/ (accessed on 4 October 2022). The dataset contained 1097 observations. Mortality associated with a risk factor is simply the estimated number of deaths associated with each of the 30 risk factors. These 30 risk factors are listed in Table 1.

### 2.2. Analysis Procedure

We treated LCM as the dependent variable and the 30 risk factors as independent variables in the analysis. We first computed the crude mortality rate for each variable in each of the 100 countries. The crude rate is the rate between the number of deaths associated with each risk factor divided by the average population size in each country from 2006 to 2016. We then classified the countries into five risk levels for each variable using quintiles. From the lowest quintile to the highest quintile, all 100 countries were classified into very low risk, low risk, medium risk, high risk, and very high risk in LCM, and in each of the other 30 variables. Figure 1 shows the distribution of LCM risk levels of these 100 countries in the world. Second, we performed analyses using the fuzzy inference modeling (FIM) and the random forest modelling (RFM) shown in Figure 2. In FIM, we implemented analysis using the Analytical Hierarchical Process (AHP), the RIDIT analysis, and the Chi-square analysis to search for the optimal lattice degree based on nearness. Third, we conducted the RFM. Fourth, we selected the optimal weighting group based on the results of the optimal lattice degree on nearness and compared the results from RFM. Fifth, we determined the similarity and dissimilarity in the comparison of the two approaches. Figure 2 illustrates the overall analysis framework. We provided a brief description of the fuzzy inference modeling below. Details of the RFM can be found in the article by Du and colleagues [19].

We used ArcGIS Pro 3.0 to create Figure 1 based on the World Map with Polyconic Projection with Meridional Interval on Same Parallel Decrease Away from Central Meridian by Equal Difference. ArcGIS Pro is a desktop GIS software developed by Esri, which replaces their ArcMap software generation. The product was announced as part of Esri’s ArcGIS 10.3 release. ArcGIS Pro is notable in having a 64-bit architecture, combined 2-D and 3-D support, ArcGIS Online integration and Python 3 support.

### 2.3. Fuzzy Inference Methods

The FIM is based on fuzzy logic that is used to make decisions on imprecise information. Since fuzzy logic originates from fuzzy set theory where reasoning is approximate, the fuzzy inference is used in the field of anomaly detection so that all variables are viewed as fuzzy variables [20]. The strengths of these methods include its capacity of modeling non-linearity efficiently, segregating normal and anomalous samples, and better predicting the inconsistencies [21]. The application of these methods in this study can be considered as the auxiliary validation of machine learning-based medical anomaly detection that is related to the purpose of prediction and diagnosis, in addition to the medical data analyzed by machine learning. The procedure of fuzzy inference used in this study includes the 10 steps listed below [22].

(1). Filter out unrelated variables and accept Table 1 to establish the LCM index system. In this research, we selected the 30 risk factors as the variables related to LCM.

(2). Establish all factors as vector-matrix (U) and use the five risk levels of LCM as the five classes in the analysis (V).

(3). Generate the fuzzy similar matrix (R) among the 100 countries using the product of U times V based on the formula below.
(1)R=μ1μ2⋮μm×v11v12v1nv21v22v2
n⋮⋮⋯⋮vm1vm2vmn

(4). Implement the Chi-square test, generate chi-square value and weight, and normalize the weights to obtain the first set of weights-A1.

(5). Perform RIDIT analysis to obtain the second set of weights-A2. RIDIT values are generally based on the observed distribution of a response variable for a specified set of individuals [23]. This approach is very closely related to distribution-free methods based on ranks such as the Wilcoxon Test [24]. RIDIT possesses two very important properties. First, it assigns a rank value to each class proportional to the relative frequency of observations in that class. Second, it standardizes the rank values to vary between 0 and 1. The latter property eliminates the problem of variation in the relative positions with respect to the number of ranks. RIDIT technique appears to suppress the differences in distributional shape [25].

(6). Perform analysis using the AHP method to obtain the third set of weights-A3.

(7). Use B1 = A1 × R, B2 = A2 × R, B3 = A3 × R to obtain the values of B1, B2, and B3, and then normalize these values.

(8). Compute the lattice degree of nearness σ using Equation (2) below based on the original weights C.
(2)$$\sigma =\frac{1}{2}\left[B_{i}⊗C+\left(1-B_{i}⊙C\right)\right]$$
where *B_i_*⊗*C* = $\begin{array}{c} \;\vee \\ \chi \in v \end{array}($C(x)$\wedge B\left(x\right))$ and *B_i_*⊙*C* = $\begin{array}{c} \;\wedge \\ \chi \in v \end{array}$(C(x)$\vee B\left(x\right))$, i = 1, 2, 3. 

(9). Obtain an optimal lattice degree of nearness using fuzzy pattern recognition and
(3)$$\mathrm{the}\;\mathrm{formula}:\;\sigma_{optimal}=max\left(\sigma_{1},\sigma_{2},\sigma_{3}\right)$$

(10). Select the optimal weight group based on the results from the three analytic approaches.

## 3. Results

### 3.1. Fuzzy Inference Modeling

#### 3.1.1. Chi-Square Analysis

Initially, 30 variables were used and passed the chi-square test. We used disease burden as an example in Table 2. 

**H_0_:** 
*disease burden with very low, low, middle, high, and very high risk are independent from the LCM rate,*


**H_α_:** 
*they are dependent.*


We calculated the observed and expected values to obtain five levels of disease burden mortality rates and examine the Chi-square test. The *p*-value was less than 0.001 after we compared observed results with expected results, leading to the rejection of H_0_. We hence concluded that different risk levels of LCM caused by disease burden were not independent of each other (H_a_). Once the disease burden passed the test, we computed the χ^2^ values of the disease burden to standardize the χ^2^ weights, which were shown in Table 3.

#### 3.1.2. RIDIT Analysis

First, we determined the frequencies of the five levels associated with the 30 risk factors. The frequencies in the five levels from very low risk through very high risk are 219, 221, 217, 221, and 219 among the 1097 values. Next, we calculated the mean of each risk level. These five means are 109.5, 110.5, 108.5, 110.5 109.5, respectively. Third, we computed the RIDIT value related to each of the five levels. These values are 0.0998, 0.3004, 0.5000, 0.6996, and 0.9002 from the lowest risk level to the highest risk level (Table 4). In the fourth and final step, we calculated the RIDIT values of the 30 risk factors and standardized weights shown in Table 5.

#### 3.1.3. AHP Analysis

We used the Delphi method in Table 6 to investigate the roles of the 30 independent variables in LCM and produced the AHP weights for LCM in Table 7. In the results, living environment qualities such as outdoor air pollution and air pollution were classified as the first class. Smoking and secondhand smoke related to human behavior were viewed as the second class. Diet health of patients was the third-class, including diet high in sodium, low bone mineral density, diet low in whole grains, and diet low in vegetables and fruits. Patient physical and mental health indices (e.g., disease burden) were assigned as the fourth class. Childhood health, such as low birth weight due to short gestation, was assigned as the class with the least impact. The results of the AHP analysis show the importance of environmental variables on LCM (Table 7).

#### 3.1.4. Overall Results of Fuzzy Inference Modeling

Because the lattice degree of nearness from the Chi-square method has the highest value of 69.36% (Table 8), the Chi-square method is considered the best weight method.

### 3.2. Results of the RFM 

We utilized the free application of Google Colab to perform the RFM. Google Colab uses Python 3.6 and allows researchers to share codes (Figure 3). The tree structure of the RFM is illustrated in Figure 4, and the results are given in Figure 5. In Figure 3, the maximum tree depth was 5, the minimum number of cases in the parent node was 10, and the minimum number of cases in a child node was 5. We obtained 20 nodes, 10 terminal nodes, and 5 levels of depth of the tree. The predicted accuracy of LCM was 96.17%. The tree structure of LCM in Figure 3 broke down the factor of No access to handwashing facility (NHF) by disease burden of 680 observations into two parts of 532 samples and 148 samples. A total of 663 observations were selected as training data and 434 observations were used as testing data. The results of the RFM modeling are shown in Figure 5. The top 7 risk factors associated with LCM are Smoking, Low physical activity (LPA), Child wasting (CW), Tuberculosis (TB), Low birth weight due to short gestation (LBW), Iron deficiency (IDY), and Diet low in nuts and seeds (DLN).

## 4. Discussion

This research took advantage of empirical analysis to compare the importance of lung cancer’s impact factors. Table 9 is a summary of the analysis results for the random forest modeling (RFM) and fuzzy inference modeling (FIM). The top 10 risk factors detected by each method are highlighted in yellow in Table 9. The top 10 risk factors identified by RFM are: smoking, low physical activity, child wasting, low birth weight due to short gestation, iron deficiency, diet low in nuts and seeds, vitamin A deficiency, low bone mineral density, air pollution, and diet high in sodium. Since smoking is a well-known risk factor associated with lung cancer [26,27,28,29,30], the results from RFM appear to be more meaningful compared with the results from FIM. The RFM findings that shed light on the smoking habit are highly superior to environmental factors, which is the first killer of lung cancer, pregnancy, and heart disease [31]. The main reason is related to the immune system’s impairment in the recruitment of white cells that release free radicals to kill off the pathogens. These free radicals could provoke an inflammatory overload when combined with those in cigarette smoke, stimulating the activated leukocytes that emit an array of cytokines, resulting in the generation of more inflammatory cells [32]. Meanwhile, the RFM results are based on the machine-learning bagging algorithm and use the ensemble learning technique [33]. It created as many decision trees as possible on the subset of the data and congregated the output of all decision trees. It reduced overfitting problems in decision trees and variances so that it substantially improves the accuracy in the terminated comparison. Importantly, the RFM evidence portrays that the LCM research belongs to machine learning-based medical anomaly detection that aims to predict and diagnose illnesses [34]. The RFM, therefore, is a generally advanced application of emergent disease detection.

The fuzzy inference provided an effective and quantitative weighting method to search the primary impact factors on LCM. On the one hand, its strength is capable of modeling non-linearity efficiently, segregating normal and anomalous samples, and better predicting the inconsistencies [35]. On the other hand, the FIM outcomes strengthened the RFM detections. Albeit the results from FIM are mixed, smoking was identified to be the third most important risk factor by both the AHP analysis and the Chi-square test, in accordance with the RFM outcome. Smoking, in the FIM, was not picked up by RIDIT analysis as an important risk factor, implying that the validity of the RIDIT analysis for the data used in this study is problematic. However, it does not have an impact on the FIM results. This is because the most optimal weight group was computed by the Chi-square test. The air pollution and outdoor air pollution were detected as the top two most important risk factors by AHP analysis, which were verified by Turner and colleagues, who applied and compared the association between outdoor air pollution and lung cancer to account for the global spatial variability of lung cancer [5]. These findings and other risk factors identified by AHP analysis warrant additional research. It is also important to note that four risk factors, smoking, low physical activity, child wasting, and air pollution, are the common risk factors identified by FRM and two other methods from FIM. Additional research is needed to examine the association between these four risk factors and lung cancer mortality. 

Most importantly, the difference between FIM and RFM draws focus to the Chi-square results and RFM results. This could be due to the weighing discrepancies of environment, nutrition, diet and sex. Apart from the common factors, there are five different factors that should be noticed in the top 10 factors of both results. Iron deficiency, diet low in nuts and seeds, low bone mineral density, air pollution, diet high in sodium in the RFM results were concentrated in a poorly balanced nutrition, except for air pollution. Malnutrition caused 35% of the incidences of cancer worldwide, estimated by the World Cancer Research Fund (WCRF) Report 2007 [36]. Air pollution leading to lung cancer is reported by the Lancet October 2022 [37]. The reason is that air pollution stimulated inactive cells with cancer-causing mutations to generate tumors. Simultaneously, the Chi-square findings depicted diet low in vegetables, child stunting, drug use, unsafe water source, and secondhand smoke as the five carcinogenic factors. Diet low in vegetables belongs to diet risk factors. Child stunting is chronic malnutrition, as the same as child wasting. Diet and nutrition, as two of modifiable lifestyle factors, were associated with reduced total cancer-specific mortality, updated by the WCRF and the American Institute for Cancer Research (AICR) (2018) [38,39,40,41]. Drug use was positively correlated with sexual behaviors [42], impacting on individual HIV-infection, ultimately resulting in the increased risk of developing lung cancer for the general population [43]. Unsafe water source is present in various food products, including mutagenic and carcinogenic compounds [36]. Secondhand smoke is strongly associated with small cell lung cancer [44], causing a 25% increased risk of lung cancer for non-smokers (American Cancer Society Report). Indeed, the root of the difference between the two results is the biases of the two methods. Due to overfitting, RFM results might have unacceptably high variance and consequently poor predictions on unseen data. FIM results depend on the lattice degree of nearness, which might from subjective judgment by experts. 

In addition to smoking, this study suggests that future research should examine risk factors such as low physical activity, child wasting, low birth weight due to short gestation, iron deficiency, diet low in nuts and seeds, vitamin A deficiency, low bone mineral density, air pollution, and diet high in sodium. Despite the fact that this study established two robust models to classify LCM to determine the most sensitive impact factors, some limitation should be noted. First, albeit two models in the application of LCM are novel for pinpointing etiology and pathogenesis on LCM, overfitting in the RFM model might exist so that outcomes are changed. Second, the scale of this research is too big to model spatial-temporal regression. The scale might be narrowed down in future works to make models more robust. Finally, with the advent of the Big Data Era and the development of data mining techniques, deep learning-based medical anomaly detection draws more attention [19]. Some updated Artificial Intelligence (AI) algorithms such as convolution neural networks might improve the model’s accuracy in future research [45].

## 5. Conclusions

This study demonstrates the feasibility of using Fuzzy Inference Modeling (FIM) and Random Forest Modeling (RFM) approaches to identify potential risk factors associated with Lung Cancer Mortality (LCM). The approaches may be useful and effective in exploring the association between a disease and its potential risk factors involving the analysis of large datasets. Future research efforts should expand the research to other diseases and their possible risk factors. In addition, further research is needed examine the effectiveness and difference of FIM and RFM in this type of analysis. 

## Figures and Tables

**Figure 1 ijerph-19-14161-f001:**
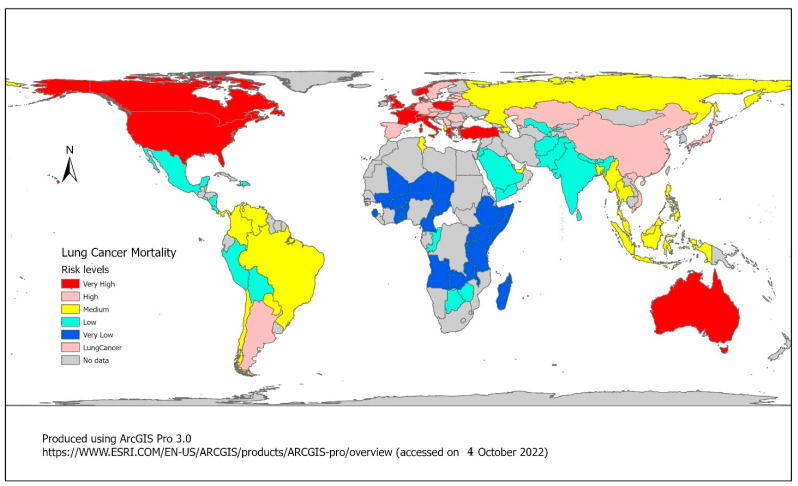
Risk levels of LCM in 100 countries from 2006 to 2016.

**Figure 2 ijerph-19-14161-f002:**
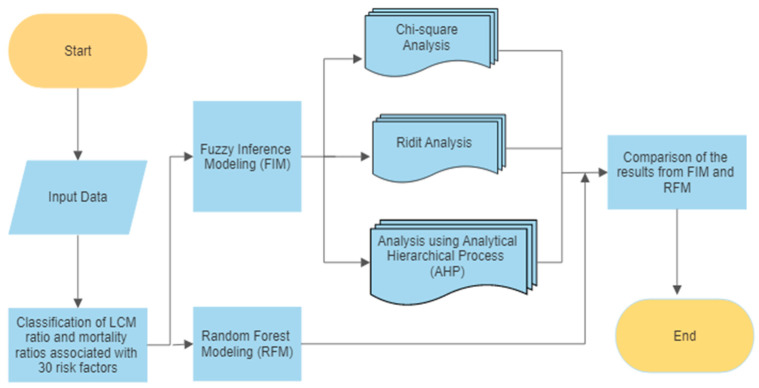
Analysis framework.

**Figure 3 ijerph-19-14161-f003:**
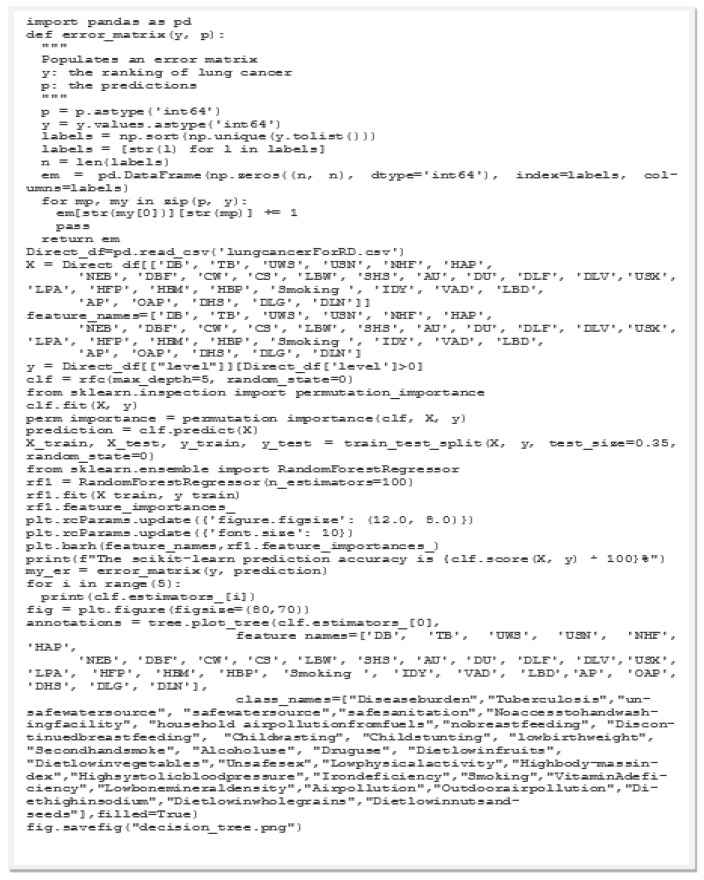
Python codes used in the FRM.

**Figure 4 ijerph-19-14161-f004:**
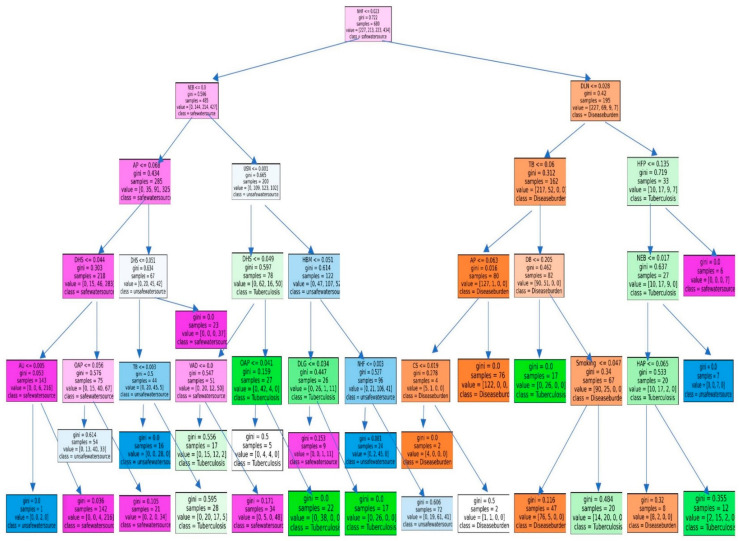
The tree structure of the RFM.

**Figure 5 ijerph-19-14161-f005:**
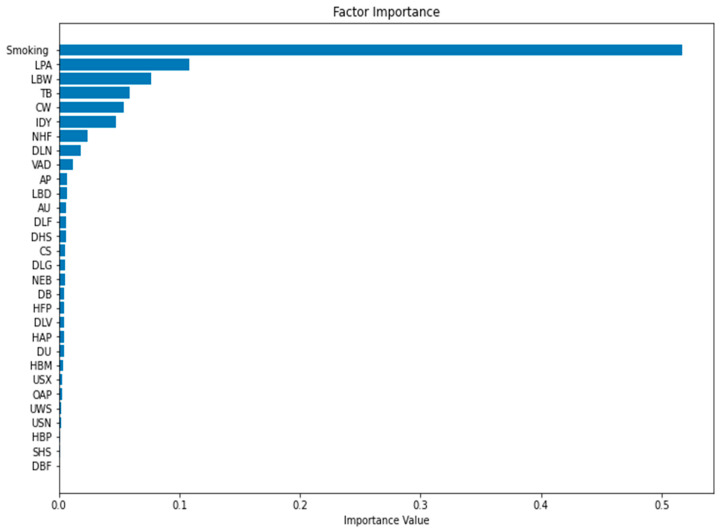
Importance of different risk factors determined by the FRM.

**Table 1 ijerph-19-14161-t001:** List of risk factors of a country used as independent variables in the analyses.

NO.	Variable Name	Acronym	Description
1	Disease burden	DB	Mortality rate from disease burden
2	Tuberculosis	TB	Mortality rate from Tuberculosis
3	Unsafe water source	UWS	Mortality rate from unsafe water source
4	Unsafe sanitation	USN	Mortality rate from unsafe sanitation
5	No access to handwashing facility	NHF	Mortality rate from no access to handwashing facility
6	Household air pollution from solid fuels	HAP	Mortality rate from household air pollution from solid fuel
7	Non-exclusive breastfeeding	NEB	Mortality rate from non-exclusive breastfeeding
8	Discontinued breastfeeding	DBF	Mortality rate from discontinued breastfeeding
9	Child wasting	CW	Mortality rate from child wasting
10	Child stunting	CS	Mortality rate from child stunting
11	Low birth weight due to short gestation	LBW	Mortality rate from low birth weight due to short gestation
12	Secondhand smoke	SHS	Mortality rate from secondhand smoke
13	Alcohol use	AU	Mortality rate from alcohol use
14	Drug use	DU	Mortality rate from drug use
15	Diet low in fruits	DLF	Mortality rate from diet low in fruits
16	Diet low in vegetables	DLV	Mortality rate from diet low in vegetable
17	Unsafe sex	USX	Mortality rate from unsafe sex of a country
18	Low physical activity	LPA	Mortality rate from low physical activity
19	High fasting plasma glucose	HFP	Mortality rate from high fasting plasma glucose
20	High body-mass index	HBM	Mortality rate from high body-mass index
21	High systolic blood pressure	HBP	Mortality rate from high systolic blood pressure
22	Iron deficiency	IDY	Mortality rate from iron deficiency
23	Smoking		Mortality rate from smoking
24	Vitamin A deficiency	VAD	Mortality rate from Vitamin A deficiency
25	Low bone mineral density	LBD	Mortality rate from low bone mineral density
26	Air pollution	AP	Mortality rate from air pollution
27	Outdoor air pollution	OAP	Mortality rate from outdoor air pollution
28	Diet high in sodium	DHS	Mortality rate from diet high in sodium
29	Diet low in whole grains	DLG	Mortality rate from diet low in whole grains
30	Diet low in nuts and seeds	DLN	Mortality rate from diet low in nuts and seeds

**Table 2 ijerph-19-14161-t002:** Process of the Chi-square analysis.

**Observed Values**						
variable	very low risk	low risk	medium risk	high risk	very high risk	total rate
DB	7.199	31.0088	63.8118	109.3561	169.6175	380.9932
non-DB	49.1659	109.6643	156.3428	188.1021	212.7317	716.0068
total	56.3649	140.6731	220.1546	297.4582	382.3492	1097
**Expected Values**						
variable	very low risk	low risk	medium risk	high risk	very high risk	total
DB	19.5758	48.8564	76.4607	103.3086	132.7917	380.9932
non-DB	36.7891	91.8167	143.6939	194.1496	249.5575	716.0068
total	56.3649	140.6731	220.1546	297.4582	382.3492	1097
chi-square test			<0.001			
**χ^2^ Values**						
variable	very low risk	low risk	medium risk	high risk	very high risk	total
DB	7.8252	6.5199	2.0925	0.3540	10.2125	27.0041
non-DB	4.1639	3.4693	1.1134	0.1884	5.4342	14.3691
Total	11.9891	9.9891	3.2059	0.5424	15.6467	41.3733

**Table 3 ijerph-19-14161-t003:** Results of Chi-square analysis.

Variable	χ^2^ Value	χ^2^ Weight
Disease burden	41.37	0.0030
Tuberculosis	516.68	0.0372
Unsafe water source	523.82	0.0377
Unsafe sanitation	533.08	0.0384
No access to handwashing facility	532.60	0.0383
Household air pollution from solid fuels	497.60	0.0358
Non-exclusive breastfeeding	553.70	0.0399
Discontinued breastfeeding	581.37	0.0418
Child wasting	646.50	0.0465
Child stunting	545.02	0.0392
Low birth weight due to short gestation	581.04	0.0418
Secondhand smoke	522.60	0.0376
Alcohol use	451.03	0.0325
Drug use	531.80	0.0383
Diet low in fruits	491.40	0.0354
Diet low in vegetables	506.60	0.0365
Unsafe sex	449.78	0.0324
Low physical activity	514.90	0.0371
High fasting plasma glucose	246.80	0.0178
High body-mass index	331.50	0.0239
High systolic blood pressure	160.50	0.0116
Iron deficiency	281.30	0.0202
Smoking	586.03	0.0422
Vitamin A deficiency	595.18	0.0428
Low bone mineral density	470.50	0.0339
Air pollution	370.06	0.0266
Outdoor air pollution	453.13	0.0326
Diet high in sodium	473.13	0.0341
Diet low in whole grains	429.40	0.0309
Diet low in nuts and seeds	475.24	0.0342
Total	13,893.66	1.0000

**Table 4 ijerph-19-14161-t004:** Process of RIDIT analysis.

Level	(1) Frequency	(2) Average	(3) Cumulative	(4) Sum	RIDIT Value
				(2) + (3)	(4)/Total Ratio
Very low risk	219	109.5	0	109.5	0.0998
Low risk	221	110.5	219	329.5	0.3004
Medium risk	217	108.5	440	548.5	0.5000
High risk	221	110.5	657	767.5	0.6996
very high risk	219	109.5	878	987.5	0.9002
Total ratio	1097				

**Table 5 ijerph-19-14161-t005:** Results of RIDIT analysis.

Variable	RIDIT Value	RIDIT Weight
Disease burden	0.247	0.244
Tuberculosis	0.018	0.018
Unsafe water source	0.020	0.020
Unsafe sanitation	0.015	0.015
No access to handwashing facility	0.014	0.014
Household air pollution from solid fuels	0.024	0.024
Non-exclusive breastfeeding	0.004	0.004
Discontinued breastfeeding	0.001	0.000
Child wasting	0.028	0.028
Child stunting	0.006	0.006
Low birth weight due to short gestation	0.021	0.021
Secondhand smoke	0.011	0.011
Alcohol use	0.034	0.034
Drug use	0.006	0.006
Diet low in fruits	0.022	0.022
Diet low in vegetables	0.015	0.015
Unsafe sex	0.041	0.041
Low physical activity	0.014	0.014
High fasting plasma glucose	0.076	0.076
High body-mass index	0.057	0.057
High systolic blood pressure	0.107	0.106
Iron deficiency	0.072	0.071
Smoking	0.001	0.001
Vitamin A deficiency	0.007	0.007
Low bone mineral density	0.003	0.003
Air pollution	0.043	0.043
Outdoor air pollution	0.027	0.026
Diet high in sodium	0.024	0.024
Diet low in whole grains	0.032	0.032
Diet low in nuts and seeds	0.022	0.022
Total	1.012	1.000

**Table 6 ijerph-19-14161-t006:** Results of the Delphi method.

Variable	DB	TB	UWS	USN	NHF	HAP	NEB	DBF	CW	CS	LBW	SHS	AU	DU	DLF	DLV	USX	LPA	HFP	HBM	HBP	Smoking	IDY	VAD	LBD	AP	OAP	DHS	DLG	DLN
DB	1.00	0.20	0.14	0.09	0.50	0.13	0.25	0.33	0.50	1.00	3.33	0.11	1.00	3.33	0.33	0.17	0.17	3.33	3.33	3.33	3.33	0.14	3.33	3.33	0.25	0.20	0.14	0.33	1.00	0.33
TB	5.00	1.00	0.11	0.10	0.50	0.25	1.00	1.00	2.00	2.00	3.33	0.13	1.00	1.00	2.00	2.00	2.00	2.00	2.00	2.00	2.00	0.20	2.00	2.00	0.25	0.20	0.14	0.33	0.33	0.33
UWS	7.00	9.00	1.00	8.00	8.00	8.00	1.00	1.00	1.00	1.00	1.00	0.17	0.50	0.50	1.00	1.00	1.00	1.00	1.00	1.00	1.00	0.11	1.00	1.00	0.25	0.14	0.10	0.33	1.00	0.33
USN	11.00	10.00	1.00	1.00	2.00	0.50	1.43	1.43	1.43	1.43	1.43	0.20	0.33	0.33	0.50	0.50	0.50	0.50	0.50	0.50	0.50	0.13	0.33	0.33	0.25	0.13	0.10	0.33	0.14	0.33
NHF	2.00	2.00	1.50	0.50	1.00	0.50	1.00	1.00	1.00	1.00	1.00	0.11	0.50	1.00	0.20	0.20	0.20	0.20	1.00	1.00	1.00	0.11	0.33	1.00	0.25	0.14	0.10	0.33	2.00	2.00
HAP	8.00	4.00	3.00	2.00	2.00	1.00	0.33	0.33	0.50	1.00	2.00	0.14	1.00	2.00	0.50	1.00	0.33	0.50	1.00	0.33	0.20	0.11	2.00	2.00	0.25	0.20	0.11	0.33	0.33	2.00
NEB	4.00	1.00	1.00	0.70	1.00	3.00	1.00	1.00	1.00	1.00	1.00	0.20	0.50	1.00	0.50	0.50	1.00	1.00	1.00	1.00	1.00	0.11	1.00	1.00	0.25	0.14	0.11	0.33	0.33	2.00
DBF	3.00	1.00	1.00	0.70	1.00	3.00	1.00	1.00	1.00	1.00	1.00	0.14	0.50	1.00	0.50	0.50	2.00	2.00	2.00	2.00	1.00	0.11	1.00	1.00	0.25	0.14	0.11	0.33	0.50	1.43
CW	2.00	0.50	1.00	0.70	1.00	2.00	1.00	1.00	1.00	1.00	1.00	0.14	0.50	1.00	0.50	0.50	0.11	0.20	0.20	0.33	1.00	0.11	1.00	1.00	0.25	0.14	0.11	0.33	0.50	1.43
CS	1.00	0.50	1.00	0.70	1.00	1.00	1.00	1.00	1.00	1.00	1.00	0.14	0.50	1.00	0.50	0.50	0.14	0.14	0.14	0.14	0.14	0.11	1.00	1.00	0.25	0.14	0.11	0.33	0.50	1.43
LBW	0.30	0.30	1.00	0.70	1.00	0.50	1.00	1.00	1.00	1.00	1.00	0.11	0.25	0.33	0.25	0.25	0.11	0.11	0.11	0.11	0.11	0.09	0.33	0.33	0.25	0.11	0.09	0.33	0.33	0.33
SHS	9.00	8.00	6.00	5.00	9.00	7.00	5.00	7.00	7.00	7.00	9.00	1.00	2.00	10.00	3.33	3.33	10.00	10.00	10.00	10.00	10.00	0.50	10.00	10.00	0.25	1.00	0.50	0.33	0.33	10.00
AU	1.00	1.00	2.00	3.00	2.00	1.00	2.00	2.00	2.00	2.00	4.00	0.50	1.00	1.00	0.50	0.50	0.14	0.14	0.14	0.14	0.14	0.11	1.00	1.00	0.25	0.14	0.11	0.33	0.50	1.43
DU	0.30	1.00	2.00	3.00	1.00	0.50	1.00	1.00	1.00	1.00	3.00	0.10	1.00	1.00	0.50	0.50	0.14	0.14	0.14	0.14	0.14	0.11	1.00	1.00	0.25	0.14	0.11	0.33	0.50	1.43
DLF	3.00	0.50	1.00	2.00	5.00	2.00	2.00	2.00	2.00	2.00	4.00	0.30	2.00	2.00	1.00	1.00	2.00	2.00	2.00	2.00	2.00	0.11	1.00	1.00	1.00	0.11	0.09	0.33	1.00	1.00
DLV	6.00	0.50	1.00	2.00	5.00	1.00	2.00	2.00	2.00	2.00	4.00	0.30	2.00	2.00	1.00	1.00	1.00	2.00	2.00	2.00	2.00	0.11	1.00	1.00	1.00	0.11	0.09	0.33	1.00	1.00
USX	6.00	0.50	1.00	2.00	5.00	3.00	1.00	0.50	9.00	7.00	9.00	0.10	7.00	7.00	0.50	1.00	1.00	1.00	1.00	1.00	1.00	0.09	0.50	0.50	0.50	0.09	0.08	0.33	1.00	1.00
LPA	0.30	0.50	1.00	2.00	5.00	2.00	1.00	0.50	5.00	7.00	9.00	0.10	7.00	7.00	0.50	0.50	1.00	1.00	1.00	1.00	1.00	0.09	0.50	0.50	0.50	0.09	0.08	0.33	1.00	1.00
HFP	0.30	0.50	1.00	2.00	1.00	1.00	1.00	0.50	5.00	7.00	9.00	0.10	7.00	7.00	0.50	0.50	1.00	1.00	1.00	1.00	1.00	0.09	0.50	0.50	0.50	0.09	0.08	0.33	1.00	1.00
HBM	0.30	0.50	1.00	2.00	1.00	3.00	1.00	0.50	3.00	7.00	9.00	0.10	7.00	7.00	0.50	0.50	1.00	1.00	1.00	1.00	1.00	0.09	0.50	0.50	0.50	0.09	0.08	0.33	1.00	1.00
HBP	0.30	0.50	1.00	2.00	1.00	5.00	1.00	1.00	1.00	7.00	9.00	0.10	7.00	7.00	0.50	0.50	1.00	1.00	1.00	1.00	1.00	0.09	0.50	0.50	0.50	0.09	0.08	0.33	1.00	1.00
Smoking	7.00	5.00	9.00	8.00	9.00	9.00	9.00	9.00	9.00	9.00	11.00	2.00	9.00	9.00	9.00	9.00	11.00	11.00	11.00	11.00	11.00	1.00	20.00	20.00	20.00	1.30	1.00	2.50	2.00	2.00
IDY	0.30	0.50	1.00	3.00	3.00	0.50	1.00	1.00	1.00	1.00	3.00	0.10	1.00	1.00	1.00	1.00	2.00	2.00	2.00	2.00	2.00	0.05	1.00	0.50	0.50	0.09	0.08	0.33	1.00	1.00
VAD	0.30	0.50	1.00	3.00	1.00	0.50	1.00	1.00	1.00	1.00	3.00	0.10	1.00	1.00	1.00	1.00	2.00	2.00	2.00	2.00	2.00	0.05	2.00	1.00	0.50	0.09	0.08	0.33	1.00	1.00
LBD	4.00	4.00	4.00	4.00	4.00	4.00	4.00	4.00	4.00	4.00	4.00	4.00	4.00	4.00	1.00	1.00	2.00	2.00	2.00	2.00	2.00	0.05	2.00	2.00	1.00	0.09	0.08	0.33	1.00	1.00
AP	5.00	5.00	7.00	8.00	7.00	5.00	7.00	7.00	7.00	7.00	9.00	1.00	7.00	7.00	9.00	9.00	11.00	11.00	11.00	11.00	11.00	0.77	11.00	11.00	11.00	1.00	1.00	20.00	100.00	100.00
OAP	7.00	7.00	10.00	10.00	10.00	9.00	9.00	9.00	9.00	9.00	11.00	2.00	9.00	9.00	11.00	11.00	13.00	13.00	13.00	13.00	13.00	1.00	13.00	13.00	13.00	1.00	1.00	20.00	100.00	100.00
DHS	3.00	3.00	3.00	3.00	3.00	3.00	3.00	3.00	3.00	3.00	3.00	3.00	3.00	3.00	3.00	3.00	3.00	3.00	3.00	3.00	3.00	0.40	3.00	3.00	3.00	0.05	0.05	1.00	1.11	1.11
DLG	1.00	3.00	1.00	7.00	0.50	3.00	3.00	2.00	2.00	2.00	3.00	3.00	2.00	2.00	1.00	1.00	1.00	1.00	1.00	1.00	1.00	0.50	1.00	1.00	1.00	0.01	0.01	0.90	1.00	1.00
DLN	3.00	3.00	3.00	3.00	0.50	0.50	0.50	0.70	0.70	0.70	3.00	0.10	0.70	0.70	1.00	1.00	1.00	1.00	1.00	1.00	1.00	0.50	1.00	1.00	1.00	0.01	0.01	0.90	1.00	1.00
Sum	101.40	74.00	67.75	89.19	92.00	79.88	64.51	63.80	85.13	98.13	136.10	19.60	86.28	100.20	52.62	53.45	71.85	76.27	77.57	77.04	76.57	7.06	83.83	83.00	59.00	7.30	5.82	53.30	223.42	239.92

**Table 7 ijerph-19-14161-t007:** Results of the AHP Analysis.

Variables	Priority Vector	Weights
Disease burden	0.0159	0.0159
Tuberculosis	0.0184	0.0184
Unsafe water source	0.0253	0.0253
Unsafe sanitation	0.0175	0.0175
No access to handwashing facility	0.0107	0.0107
Household air pollution from solid fuels	0.0168	0.0168
Non-exclusive breastfeeding	0.0131	0.0131
Discontinued breastfeeding	0.0144	0.0144
Child wasting	0.0103	0.0103
Child stunting	0.0090	0.0090
Low birth weight due to short gestation	0.0067	0.0067
Secondhand smoke	0.0827	0.0827
Alcohol use	0.0143	0.0143
Drug use	0.0108	0.0108
Diet low in fruits	0.0206	0.0206
Diet low in vegetables	0.0207	0.0207
Unsafe sex	0.0262	0.0262
Low physical activity	0.0220	0.0220
High fasting plasma glucose	0.0202	0.0202
High body-mass index	0.0202	0.0202
High systolic blood pressure	0.0205	0.0205
Iron deficiency	0.1288	0.1288
Smoking	0.0148	0.0148
Vitamin A deficiency	0.0146	0.0146
Low bone mineral density	0.0368	0.0368
Air pollution	0.1420	0.1420
Outdoor air pollution	0.1665	0.1665
Diet high in sodium	0.0384	0.0384
Diet low in whole grains	0.0257	0.0257
Diet low in nuts and seeds	0.0160	0.0160

**Table 8 ijerph-19-14161-t008:** Lattice Degree on Nearness.

Method	Original Weights C	C_11_	C_12_	C_13_	C_14_	C_15_	Fuzzy Inference	Lattice Degree of Nearness
		2.92%	7.18%	15.41%	30.66%	43.83%		σ
Chi-square	*B*1⊗*C*	2.92%	7.18%	14.48%	23.44%	43.83%	43.83%	
	*B*1⊙*C*	5.11%	9.52%	15.41%	30.66%	47.44%	5.11%	
	*C∙B*1							69.36%
RIDIT Analysis	*B*2⊗*C*	2.92%	7.18%	15.41%	26.65%	42.35%	42.35%	
	*B*2⊙*C*	3.98%	9.97%	17.05%	30.66%	43.83%	3.98%	
	*C∙B*2							69.18%
AHP	*B*3⊗*C*	2.92%	7.18%	15.41%	24.12%	40.16%	40.16%	
	*B*3⊙*C*	6.74%	11.95%	17.02%	30.66%	43.83%	6.74%	
	*C∙B*3							66.71%

**Table 9 ijerph-19-14161-t009:** Comparison of the analysis results from different methods.

Main Factors	RFM Ranking	AHP Weight	AHP Ranking	Chi-Square Weight	Chi-Square Ranking	RIDIT Weight	RIDIT Ranking
Smoking	1	0.015	3	0.042	3	0.002	24
Low physical activity	2	0.022	10	0.037	9	0.013	18
Child Wasting	3	0.010	22	0.047	1	0.032	8
Low birth weight due to short gestation	4	0.007	24	0.042	4	0.023	13
Iron deficiency	5	0.129	18	0.020	22	0.067	3
Diet low in nuts and seeds	6	0.016	16	0.034	13	0.021	16
Vitamin A deficiency	7	0.015	19	0.043	2	0.008	20
Low bone mineral density	8	0.037	6	0.034	15	0.003	23
Air pollution	9	0.142	2	0.027	20	0.040	6
Diet high in sodium	10	0.038	5	0.034	14	0.024	12
Household air pollution from solid fuels	11	0.017	15	0.036	11	0.025	11
Diet low in fruits	12	0.021	12	0.035	12	0.021	15
Disease burden	13	0.016	17	0.003	24	0.245	1
Diet low in vegetables	14	0.021	11	0.037	10	0.014	17
Alcohol use	15	0.014	20	0.033	17	0.033	7
Drug use	16	0.011	21	0.038	6	0.006	22
High body-mass index	17	0.020	13	0.024	21	0.054	4
Child stunting	18	0.009	23	0.039	5	0.007	21
Unsafe water source	19	0.025	9	0.038	7	0.023	14
High fasting plasma glucose	20	0.020	14	0.018	23	0.070	2
Diet low in whole grains	21	0.026	8	0.031	19	0.031	9
Outdoor air pollution	22	0.167	1	0.033	16	0.025	10
Unsafe sex	23	0.026	7	0.032	18	0.048	5
Secondhand smoke	24	0.083	4	0.038	8	0.010	19

## Data Availability

Not applicable.

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
