# Peer review of "Determining Association between Lung Cancer Mortality Worldwide and Risk Factors Using Fuzzy Inference Modeling and Random Forest Modeling"

_ijerph, 2022, doi:10.3390/ijerph192114161_

Round 1

Reviewer 1 Report

Dear Authors,

Please, take these suggestions into consideration:

     1-   ABSTRACT. Please, divide the abstract into the correct sections for a research article (i.e. background, methods, results ...).

     2-     ABSTRACT. What do you mean by “low birth weight for gestation”? Did you mean “low birth weight due to short gestation” or “low birth weight for gestational age”? If so, correct accordingly. Otherwise, revise the sentence to clarify its meaning.

     3-     INTRODUCTION. Line 29. Please, replace “represents” with “has”.

     4-     Figure 4. Please, if possible, provide a sharper image.

     5-     DISCUSSION. Lines 237-240. “In spite of the fact that smoking in the FIM is not picked up by Ridit analysis as an important risk factor, implying the validity of the Ridit analysis for the data used in this study is problematic, which not impacts on the FIM consequence, thanks of the most optimal weight group is computed by the Chi-square test”. Please, rephrase this sentence to make it clearer, correcting the syntactical and lexical errors as well.

     6-     DISCUSSION. Lines 241-244 “The air pollution and outdoor air pollution are detected as the top two most importation risk factors by AHP analysis, which verified Turner and colleagues, who applied and compared association between outdoor air pollution and lung cancer to account for the global spatial variability of lung cancer”. How do you explain the difference between AHP and RFM regarding the importance of air pollution as a risk factor for LCM? Also, correct “importation” (line 242) with “important”.

     7-     DISCUSSION. Line 258 “It might be narrowed down scales in future works…” Please, correct as follows: “The scale might be narrowed down in future works…”

     8-     CONCLUSION. Line 267 “Nevertheless, there still more consideration to prevent and conquer lung cancer.” Please, correct the sentence.

Author Response

Dear Editors,

Thank you so much for giving us the opportunity to revise the manuscript (Manuscript ID: ijerph-1981602). Please extend our thanks to the two anonymous reviewers for their valuable suggestions and comments. We have reviewed these comments carefully and have made revisions accordingly.  In addition, we have responded to some comments from the reviewers when appropriate.

We revised the abstract, introduction, discussion, and conclusion.  Given below is a summary of the revisions and responses to the reviewers’ suggestions and comments.  For your reading convenience, we colored the suggestions and comments from the editors and reviewers in blue.

Sincerely,

Xiu Wu

The First Reviewer’s Comments and Suggestions for Authors

Dear Authors,

Please, take these suggestions into consideration:

We appreciate the thoughtful suggestions.

  • Please, divide the abstract into the correct sections for a research article (i.e. background, methods, results ...).

Thanks for the suggestion. We rewrote the Abstract part. Please review it in the new manuscript.

 2-     ABSTRACT. What do you mean by “low birth weight for gestation”? Did you mean “low birth weight due to short gestation” or “low birth weight for gestational age”? If so, correct accordingly. Otherwise, revise the sentence to clarify its meaning.

This is a constructive point. We appreciate your attention. We corrected it on the ABSTRACT (Line 17), Table 1, Table 3, Table 7, Table 9, Line 177, Line 198, Line 218, and Line 254 in the new manuscript.

  • Line 29. Please, replace “represents” with “has”.

Thanks for your advice. That is a good point we did not realize. We corrected it in the new manuscript of Line 31.

  • Figure 4. Please, if possible, provide a sharper image.

Thanks for your suggestions. We already have redrawn Figure 4. It is hard to further increase image resolutions.

  • Lines 237-240. “In spite of the fact that smoking in the FIM is not picked up by Ridit analysis as an important risk factor, implying the validity of the Ridit analysis for the data used in this study is problematic, which not impacts on the FIM consequence, thanks of the most optimal weight group is computed by the Chi-square test”. Please, rephrase this sentence to make it clearer, correcting the syntactical and lexical errors as well.

We appreciate the suggestions. We revised the sentence (Lines 239-242).

  • Lines 241-244 “The air pollution and outdoor air pollution are detected as the top two most importation risk factors by AHP analysis, which verified Turner and colleagues, who applied and compared association between outdoor air pollution and lung cancer to account for the global spatial variability of lung cancer”. How do you explain the difference between AHP and RFM regarding the importance of air pollution as a risk factor for LCM? Also, correct “importation” (line 242) with “important”.

It is an excellent question. The AHP results were derived from subjective judgement by experts, but RFM results were data-driven and objectively and automatically produced by machine-learning algorithms. The difference between two methods displayed the weighing discrepancy between environmental factors and congenital deficiency by potential genetic effects. We changed “importation” into “important” on Line 245.

  • Line 258 “It might be narrowed down scales in future works…” Please, correct as follows: “The scale might be narrowed down in future works…”

Thanks for the suggestion. We corrected it in the revised version of the manuscript as suggested (Lines 261-262).

     8-     CONCLUSION. Line 267 “Nevertheless, there still more consideration to prevent and conquer lung cancer.” Please, correct the sentence.

We appreciate the thoughtful suggestions. We completely rewrote the texts in the Conclusion section.

Reviewer 2 Report

The article entitled Determining Association between Lung Cancer Mortality 2 Worldwide and Risk Factors Using Fuzzy Inference Modeling 3 and Random Forest Modeling” used Fuzzy Inference Modeling (FIM) and Random Forest Modeling (RFM) approaches 14 to analyze The datasets contained estimated LCM and 15 mortality from 30 risk factors in the 100 countries over the period from 2006 to 2016. Major concerns are shown below:

1.     The Global Burden of Disease database included data from 1990 to 2019. Why do the authors only used data from 2006 to 2016?

2.     The authors should provide detailed introduction to the risk factors data. Is it individual data or estimated population data? Will it be feasible to include an example of the risk factors data?

3.     What is the underlying reason to classify the countries into five risk levels for each variable using quintiles? Are there any references to perform this kind of transformation?

4.     Results are different between Fuzzy Inference Modeling (FIM) and Random Forest Modeling (RFM) approaches. What do we know about the reason? What implications can we have based on such results? This issue should be further discussed.

Author Response

Dear Editors,

Thank you so much for giving us the opportunity to revise the manuscript (Manuscript ID: ijerph-1981602). Please extend our thanks to the two anonymous reviewers for their valuable suggestions and comments. We have reviewed these comments carefully and have made revisions accordingly.  In addition, we have responded to some comments from the reviewers when appropriate.

We revised the abstract, introduction, discussion, and conclusion.  Given below is a summary of the revisions and responses to the reviewers’ suggestions and comments.  For your reading convenience, we colored the suggestions and comments from the editors and reviewers in blue.

Sincerely,

Xiu Wu

Round 2

Reviewer 2 Report

As mentioned in the last review report, results are different between Fuzzy Inference Modeling (FIM) and Random Forest Modeling (RFM) approaches, which should be further discussed. More efforts should be made in the discussion. 

Author Response

Dear Reviewer,

Thank you so much for giving us the opportunity to revise the manuscript (Manuscript ID: ijerph-1981602). We are also grateful for your valuable suggestion. We revised the discussion.  For your reading convenience, we colored the suggestion in blue.

Sincerely,

Xiu Wu

The second reviewer’s comments:

As mentioned in the last review report, results are different between Fuzzy Inference Modeling (FIM) and Random Forest Modeling (RFM) approaches, which should be further discussed. More efforts should be made in the discussion. 

Thanks for the straightforward suggestion.  We added the following contents in the new manuscript (Line 258).

 Most importantly, the difference between FIM and RFM draws focus to the Chi-square results and RFM results. This could be due to the weighing discrepancies of environment, nutrition, diet and sex. Apart from the common factors, there are five different factors that should be noticed in the top 10 factors of both results. Iron deficiency, diet low in nuts and seeds, low bone mineral density, air pollution, diet high in sodium in the RFM results were concentrated in a poorly balanced nutrition, except for air pollution. Malnutrition caused 35% of the incidences of cancer worldwide, estimated by the World Cancer Research Fund (WCRF) Report 2007 [36]. Air pollution leading to lung cancer is reported by the Lancet October 2022 [37]. The reason is that air pollution stimulated inactive cells with cancer-causing mutations to generate tumors. Simultaneously, the Chi-square findings depicted diet low in vegetables, child stunting, drug use, unsafe water source, and secondhand smoke as the five carcinogenic factors. Diet low in vegetables belongs to diet risk factors. Child stunting is chronic malnutrition, as the same as child wasting. Diet and nutrition, as two of modifiable lifestyle factors, were associated with reduced total cancer-specific mortality, updated by the WCRF and the American Institute for Cancer Research (AICR) (2018) [38-41]. Drug use was positively correlated with sexual behaviors [42], impacting on individual HIV-infection, ultimately resulting in the increased risk of developing lung cancer for the general population [43]. Unsafe water source is present in various food products, including mutagenic and carcinogenic compounds [36]. Secondhand smoke is strongly associated with small cell lung cancer [44], causing a 25% increased risk of lung cancer for non-smokers (American Cancer Society Report).
